# PCSK9 as an Atherothrombotic Risk Factor

**DOI:** 10.3390/ijms24031966

**Published:** 2023-01-19

**Authors:** Tadeja Sotler, Miran Šebeštjen

**Affiliations:** 1Department of Cardiology, University Medical Centre Ljubljana, 1000 Ljubljana, Slovenia; 2Department of Vascular Diseases, University Medical Centre Ljubljana, 1000 Ljubljana, Slovenia; 3Faculty of Medicine, University of Ljubljana, 1000 Ljubljana, Slovenia

**Keywords:** PCSK9, LDL cholesterol, atherosclerosis, inflammation, coagulation

## Abstract

Disturbances in lipid metabolism are among the most important risk factors for atherosclerotic cardiovascular disease. Proprotein convertase subtilisin/kexin type 9 (PCSK9) is a key protein in lipid metabolism that is also involved in the production of inflammatory cytokines, endothelial dysfunction and aherosclerotic plaque development. Studies have shown a connection between PCSK9 and various indicators of inflammation. Signalling pathways that include PCSK9 play important role in the initiation and development of atherosclerotic lesions by inducing vascular inflammation. Studies so far have suggested that PCSK9 is associated with procoagulation, enhancing the development of atherosclerosis. Experimentally, it was also found that an increased concentration of PCSK9 significantly accelerated the apoptosis of endothelial cells and reduced endothelial function, which created conditions for the development of atherosclerosis. PCSK9 inhibitors can therefore improve clinical outcomes not only in a lipid-dependent manner, but also through lipid-independent pathways. The aim of our review was to shed light on the impact of PCSK9 on these factors, which are not directly related to low-density lipoprotein (LDL) cholesterol metabolism.

## 1. Introduction

Atherosclerotic cardiovascular disease is a primary cause of morbidity and mortality worldwide. It is associated with multiple risk factors. Among the most important risk factors are disturbances in lipid metabolism, in particular, in low density lipoprotein (LDL) cholesterol and lipoprotein (a) (Lp(a)). These factors are also associated with imbalances in coagulation, the fibrinolytic system and inflammation, and hence, atherosclerosis is also an inflammatory degenerative disease [1]. Emerging data have shown that proprotein convertase subtilisin/kexin type 9 (PCSK9), a key protein in lipid metabolism, is involved in the production of inflammatory cytokines, endothelial dysfunction, atherosclerotic plaque development, rupture and subsequent atherothrombosis, leading to acute cardiovascular events [2]. PCSK9 is primarily biosynthesized in the hepatocytes. After reaching the basolateral surface of the hepatocyte, it binds to low-density lipoprotein receptor (LDLR) in an autocrine effect. Subsequently, the complex composed of LDL cholesterol, LDLR and PCSK9 is internalized into hepatocytes and undergoes endocytosis and lysosomal degradation, thereby reducing LDLR on the cell membrane and raising LDL cholesterol levels [3]. In addition to the small intestine, lungs, kidneys, pancreas and brain, PCSK9 is also highly expressed in vascular endothelial cells, smooth muscle cells and macrophages [4], subsequently exerting local effects on vascular homeostasis [5]. The finding that humans without PSCK9 expression or activity possess low levels of LDL cholesterol without harmful consequences fostered the development of powerful PCSK9 inhibitors (PCSK9i). These inhibitors block the function of PCSK9 on LDLR, resulting in drastically lower levels of circulating LDL cholesterol [6]. In the general population, serum PCSK9 concentration is associated with a future risk of cardiovascular disease, even after adjustments are made for established cardiovascular disease risk factors [7]. In patients with acute coronary syndrome, PCSK9 concentrations did not appear to be a predictive risk factor for recurrent cardiovascular events within one year. However, we must consider that patients who were already treated with statins at the time of their first acute coronary syndrome had significantly higher concentrations of PCSK9 that further increased during the first year. This increase was likely also due to the fact that all the patients were treated with statins since their first event [8]. However, PCSK9 affects the atherosclerotic process not only through the LDL cholesterol metabolism, but also through its effects on inflammation, endothelial and platelet function, and the coagulation fibrinolytic system (Table 1). The aim of our review was to shed light on the impact of PCSK9 on these factors which are not directly related to LDL cholesterol metabolism. To this end, we performed a systematic review following PRISMA guidelines [9]. The flow diagram for inclusion of the studies is shown in Appendix A.

## 2. PCSK9 and Inflammation

Chronic inflammation, along with other factors such as high blood pressure, diabetes and smoking, has become the critical pathway leading to the development and progression of atherosclerotic cardiovascular disease. The inflammatory process involves the mutual cooperation of a large number of different inflammatory cells and an even greater number of inflammatory mediators [30]. An increased concentration of inflammatory and pro-inflammatory mediators is associated with an increased tendency for atherosclerotic plaques to rupture. An increased concentration of the same mediators is also associated with an imbalance in the coagulation fibrinolytic system in favour of a reduced fibrinolytic potential and, thus, the formation of acute thrombosis in a ruptured atherosclerotic bed and the resulting acute cardiovascular event [31]. Studies have shown a connection between PCSK9 and various indicators of inflammation, in particular, with hs-CRP, fibrinogen, interleukin (IL)-6 and the number of white blood cells [2] (Figure 1). Two signalling pathways have been suggested to be involved in the positive regulation of inflammatory cytokines expression and atherosclerotic lesions formation by PCSK9. The toll-like receptor 4 (TLR4)/nuclear factor-kappa B (NF-*k*B) signalling pathway has been found to be the primary pathway that mediates the PCSK9-induced expression of pro-inflammatory cytokine, and it plays an important role in the initiation and development of atherosclerotic lesions by inducing vascular inflammation [32]. The activation of the PCSK9-LOX-1 axis has also been demonstrated to participate in the PCSK9-mediated inflammatory response. During the formation of atherosclerotic plaques, circulating oxidized LDL (ox-LDL) is bound to scavenger receptors of inflammatory mediators such as LOX-1 that are located on the surface of the endothelial cells [33]. LOX-1 is the principal receptor for ox-LDL on endothelial cells and vascular smooth muscle cells. It is expressed when macrophages, smooth muscle cells and fibroblasts are exposed to ox-LDL, angiotensin II or pro-inflammatory cytokines. The activation of LOX-1 stimulates the expression of PCSK9 [10].

In a cross-sectional study that included 552 patients with coronary artery disease (CAD) and 479 controls not treated with statins, no differences in PSCK9 concentrations were found between the groups. However, when these values were balanced for accompanying factors (e.g., gender, age, diabetes and arterial hypertension), the PCSK9 values were significantly higher in the CAD patients [34]. PCSK9 concentration, which was associated with both lipoprotein values and inflammatory parameters, was associated with the extent of coronary disease in the CAD patients. The authors concluded that the effect of PCSK9 on CAD was primarily mediated by the increased concentrations of atherogenic lipids and inflammatory markers. The findings of Gencer et al. [8] are also very similar, demonstrating the relationship between the concentrations of PCSK9 and hs-CRP in 2030 patients with acute coronary syndrome. However, they also found that PCSK9 concentrations were not predictors for recurrent coronary events within one year. Hence, these studies have shown that PCSK9 is associated with the occurrence of inflammation and the development of CAD [34]. Data have shown that the activation of the endothelium may lead to the secretion of surface molecules which are taken into inflammatory cells, such as monocytes and macrophages, that subsequently migrate across the endothelium and accumulate beneath the intima. These cells release cytokines and produce a pro-inflammatory microenvironment. Lectin-like OXLDL Receptor-1 (LOX-1) binds to circulating ox-LDL on vascular smooth muscle cells, allowing monocytes and macrophages to enter the vascular stroma, which results in the formation of foam cells [33,35]. Studies have confirmed that PCSK9 enhances the infiltration of inflammatory monocytes into the vessel wall by virtue of PCSK9-LDLR interaction in plaques, which directly promotes the formation of inflammatory atherosclerotic plaques [11]. Several studies have demonstrated that PCSK9 directly promotes atherosclerotic inflammation through altering plaque composition and accelerating inflammatory monocyte infiltration and differentiation in plaques. The finding that this can occur independently of cholesterol regulation supported the notion that PCSK9i can improve clinical outcomes not only via lipid-dependent, but also via lipid-independent pathways [36]. The data revealed that functional PCSK9 inhibition reduces systemic inflammation and endothelial dysfunction by constraining leukocyte–endothelium interactions [37].

Perhaps even more important than the systemic effects of PCSK9 on inflammation is its local effect on atherosclerotic plaques. By increasing its vulnerability and, thus, the possibility of rupture, PCSK9 escalates the possibility of a resultant acute cardiovascular event. Despite the fact that the most important sources of PCSK9 are hepatocytes, the kidneys and the small intestine, we must not neglect PCSK9 that is produced by the endothelial cells, vascular smooth muscle cells and macrophages. Through LDLR, PCSK9 increases the expression of genes encoding pro-inflammatory markers such as TNF-α and IL-1β, and it decreases the expression of genes encoding anti-inflammatory markers such as IL-10 and arginase-1 [38]. PCSK9 monoclonal antibodies have no influence on hs-CRP levels, regardless of the PCSK9i type, patient characteristics, concomitant treatment or treatment duration [12]. Similarly, in patients with elevated LDL cholesterol and Lp(a) levels that were mostly already treated with statins, additional treatment with PCSK9i evolocumab did not alter the local inflammation in the arterial wall, nor did it alter systemic inflammation [13]. Contrary to this, in patients with CAD or familial hypercholesterolemia not treated with statins due to statin intolerance, treatment with alirocumab attenuated arterial wall inflammation without changing systemic hs-CRP [39]. In both studies, local inflammation was measured using 18F-fluoro-deoxyglucose positron-emission tomography/computed tomography (18F-FDG PET/CT). Arterial 18F-FDG uptake was correlated with arterial macrophage content [13]. It could be concluded that treatment with PCSK9i is effective in reducing local inflammation only in patients who have not previously been treated with statins. There could be at least two reasons for this observation: first, on one hand, statins have a beneficial effect on inflammation, both on a systemic and a local level, and PCSK9i has no additional significant effect, and second, on the other hand, statins, regardless of the type of statin or its dose, significantly increase the concentrations of PCSK9 which reduce the beneficial effects of PCSK9i. However, long term treatment with PCSK9i significantly reduced arterial FDG uptake independently of LDL cholesterol changes. The effect was more pronounced in patients with higher baseline inflammatory burdens [40]. The increased concentration of PCSK9 accelerated the uptake of lipopolysaccharides in both hepatocytes and endothelial cells, which caused increased inflammation in the vascular wall [41]. Despite the fact that both alirocumab and evolocumab significantly reduced LDL cholesterol, a significant reduction in the volume of atherosclerotic plaques, as well as their composition, was observed, and there was no reduction in hs-CRP [42,43]. Thus, there should be less cholesterol available for oxidization and fewer activated macrophages. Given that both statins and bempedoic acid reduce both LDL cholesterol and hs-CRP, we could conclude that only drugs that inhibit cholesterol synthesis, namely, bempedoic acid and statins, are able to decrease hs-CRP levels.

## 3. PCSK9 and Haemostasis

Data from experimental studies in animals have suggested that PCSK9 can modulate both primary and secondary haemostasis indirectly or directly. It does so indirectly by affecting LDL cholesterol and directly by affecting platelet activation and factor VIII plasma levels [44]. It is believed that PCSK9, in addition to regulating LDL cholesterol in the plasma, is associated with procoagulation, thus enhancing the development of atherosclerosis [45]. A previous study showed that in patients with stable CAD, the concentration of circulating PCSK9 was independently associated with the main protein of coagulation cascade, fibrinogen [46].

### 3.1. Platelet Function

Under normal physiological conditions, the adhesion of platelets to intact endothelial cells does not occur [47]. P-selectin, a cell adhesion molecule that is expressed on both platelets and endothelial cells, is the primary mediator that enables the attachments between platelets and endothelial cells [14]. The platelets’ roles in this process depend on their capacity to respond to specific receptors expressed on damaged endothelia [15]. Data support the hypothesis that circulating PCSK9 may affect the platelet activation pathway using different mechanisms (Figure 2). Dyslipidemia induces ox-LDL generation and, in turn, facilitates platelet activation by binding to scavenger receptors, including CD36 and LOX1. PCSK9-induced platelet activation involves these two receptors on a platelet’s surface [48,49]. Previous studies have revealed a mechanism wherein PCSK9 binds to CD36 on a platelet, thus activating SRC kinase, mitogen-activated protein kinase (MAPK) and extracellular signals and increasing the generation of reactive oxygen species and activating the signalling pathways downstream of CD36 [50]. Navarese et al. [51] explored platelet aggregation and the onset of major adverse cardiovascular events (MACEs) in PCSK9-REACT patients with acute coronary syndrome who were receiving prasugrel or ticagrelor and P2Y12 inhibitors and undergoing percutaneous coronary intervention (PCI). Their data showed that increased PCSK9 levels were associated with higher platelet reactivity and may be a predictor of ischemic events in such patients. A recent study, conducted on patients with CAD, proved the platelets’ abilities to store PCSK9 and release the stored PCSK9 when activated. The released PCSK9 promoted platelet aggregation, thrombus formation, monocyte chemotaxis and monocytes differentiation into macrophages and foam cells [52]. Platelets are an important factor both in the initial stages of the atherosclerotic process, which begins with endothelial damage, and in the process of atherothrombosis, which leads to an acute cardiovascular event. Activated platelets release different growth factors and adhesion molecules that play important roles in proliferation, adhesion, coagulation and proteolysis, all of which participate in the formation of atherosclerotic plaques [14]. Although LDL cholesterol is also directly involved in the activation of platelets [16], PCSK9, as one of the main molecules in the metabolism of LDL cholesterol, acts on the activation of platelets completely independently of the LDL cholesterol metabolism [50]. The same study demonstrated that PCSK9, by binding to scavenger receptor CD36 on the surface of platelets, increased their activation and in vivo thrombosis. Platelet activation and in vivo thrombosis were abolished by acetyl salicylic acid or PCSK9i. Increased PCSK9 concentrations also increased microvascular obstruction, and thus, they increased myocardial necrosis in myocardial infarction (MI). The binding of ox-LDL to the scavenger receptor CD36 triggers a similar cascade pathway as the binding of PCSK9, consequently increasing oxygen reactive species generation [17]. PCSK9 is also highly expressed in vascular smooth muscle cells and in human atherosclerotic plaques. Its expression is regulated by many pro-atherogenic mediators, such as NADPH oxidase-derived reactive oxygen species, and it results in ox-LDL formation [53]. Once internalized, ox-LDL can induce the overexpression of PCSK9, perpetrating a pro-atherogenic stimulus, as PCSK9, in turn, is able to stimulate ox-LDL formation [54]. Despite that treatment with statins lowers concentrations of ox-LDL independently of the type or dose of statin [55], ox-LDL is still present in concentrations that, by binding to receptors on the surface of platelets, cause their activation, their increased aggregation, and, thus, the patient’s tendency to atherothrombosis. The best evidence for this is a study in which treatment with PCSK9i reduced both ox-LDL concentrations and ox-LDL-induced platelet activation [56]. All these results suggest that treatment with PCSK9i may reduce the frequency of cardiovascular complications, even in patients whose LDL cholesterol levels are well-controlled by statin treatments. Given that PCSK9i reduces platelet activation both indirectly via lowering LDL cholesterol (and, consequently ox-LDL) and directly by reducing the PCSK9 concentration, it would be expedient to consider a routine determination of the PCSK9 concentrations in high-risk patients in the future. The routine measurement of PCSK9 levels in these patients could precisely identify the patients who would benefit most from treatment with PCSK9i. Treatment with alirocumab or evolocumab in patients previously treated with statins and acetyl salicylic acid resulted in decreased platelet activation and increased sensitivity to the antiplatelet action of acetyl salicylic acid [57]. To answer the question of whether it is important to reduce the formation of PCSK9 only in hepatocytes, namely, at the primary site of PCSK9 formation—or also locally—in macrophages and endothelial cells, a study comparing PCSK9i and siRNA drugs is needed. The latter drugs selectively reduce the formation of PCSK9 in hepatocytes. Hence, only a face-to face comparison can provide an answer.

### 3.2. PCSK9 and Coagulation and the Fibrinolytic System

Disturbances in coagulation and the fibrinolytic system are not only associated with an acute coronary event, they are also important in the early stages of the atherosclerotic process [58]. Thrombus formation involves the release of prothrombotic molecules such as tissue factor (TF), platelet adhesion to the vascular wall and platelet aggregation and recruitment. This process, along with the activation of the coagulation cascade, which is responsible for fibrin production at the damaged site, leads to the formation and growth of the thrombus. The coagulation process is stimulated by the damage to the vessel wall. Here, blood comes into contact with TF, which triggers the coagulation process [59]. The result of the coagulation process is the formation of thrombin, which converts soluble fibrinogen into insoluble fibrin and is the basis of a blood clot. In a normal vessel wall, cells in the tunica media and fibroblast-like cells in the adventitia synthesize TF, which initiates the extrinsic coagulation pathway. However, no TF-containing endothelial cells were found in the vessels with a normal vessel wall [60]. The synthesis of TF in endothelial cells represents the strongest procoagulant mechanism by which endothelial cells participate in homeostasis. On the other hand, cells that synthesize and store TF have been found in atherosclerotic plaques [61] (Figure 3). Scalise et al. [62] demonstrated that PCSK9 increases mRNA and the production of TF in peripheral blood mononuclear cells. Treating these cells with PCSK9i decreased the expression of both the genes and proteins for TF. Coagulation factor VIII (FVIII) plays a very important role in the amplification and propagation of the coagulation cascade and in the generation of thrombin. Increased concentrations of FVIII have proved to be a risk factor for the occurrence of venous thrombosis [63] and arterial thrombosis and CAD [20]. LDL receptor (LDLR) -related protein-1 (LRP1) is principally responsible for the removal of circulating FVIII from the plasma by lowering LDL cholesterol. Statins increase the expression of LRP1, which, together with LDLR, increase the clearance of FVII [21]. In patients with venous thromboembolism (VTE) and high plasma FVIII levels, rosuvastatin therapy significantly lowered FVIII levels compared to a placebo [19]. However, there are no data on the effect of statins on FVIII in patients with normal FVIII levels with VTE or ischemic heart disease. Paciullo et al. [18] assessed the effects of alirocumab and atorvastatin, alone or in combination, on circulating FVIII levels and on the hepatic expression of LRP1 in a mouse model of increased FVIII. Compared with the vehicle-treated animals, the FVIII levels were significantly reduced in the hypolipemic agents-treated animals. Alirocumab, alone and in combination with atorvastatin, cancelled the effect of FVIII infusion, returning the FVIII levels to those of the non-infused controls. The lowering of FVIII attained with alirocumab plus atorvastatin was significantly stronger than that produced by atorvastatin alone. Alirocumab significantly increased hepatocyte LRP1 expression compared to a placebo. The increase, although not significant, was also evident in mice treated with atorvastatin or with a combination of both drugs. This difference can be explained by the fact that statins, regardless of the type and dose, significantly increase PCSK9 levels, thereby potentially reducing the effect of alirocumab on LPR1 expression [64]. A decrease in the concentration of FVIII was detectable after 5 days, while the concentrations of total and LDL cholesterol remained unchanged during this time. Based on this finding, we can conclude that it is a pleotropic effect that does not depend on lowering the LDL cholesterol concentration. In patients with familiar hypercholesterolemia (FH), PSCK9 concentration was shown to be related to thrombin generation, which was recognized as a global indicator of haemostatic potential [65]. On the other hand, it was shown that in patients with FH who did not tolerate statins, treatment with evolocumab did not reduce the concentration of fibrinogen and D-dimer, which are otherwise much more robust indicators of increased thrombogenicity [22]. In patients with chest pain, PCSK9 concentration was negatively associated with prothrombin time, which indicates fibrinolytic activity [66], and both were associated with the incidence of MACE. Treatment with alirocumab in drug-naïve patients with hypercholesterolemia and carotid atherosclerosis decreased fibrinogen, FVII and plasminogen activator inhibitor-1 (PAI-1) antigen levels, and it thereby increased fibrinolytic activation [67]. In the previously specified subanalysis of the ODYSSEY OUTCOME study, it was shown that the incidence of VTE was associated with the initial values of Lp(a), while its reduction during alirocumab therapy was associated with a lower incidence of VTE [68]. Similar results were also obtained in the FOURIER study, where it was shown that the occurrence of VTE was associated with an elevated Lp(a) concentration. Following evolocumab therapy, the risk of VTE decreased only in the patients with decreased in Lp(a) concentrations [69].

## 4. PCSK9 and Arterial Wall Function

The endothelium is an active inner layer of a blood vessel and is indispensable for the regulation of vascular tone and the maintenance of vascular homeostasis. Its functional impairment is characterized by an imbalance between vasodilators and contracting factors. Endothelial dysfunction represents one of the first manifestations of atherosclerosis and is involved in plaque progression and atherosclerotic complications [23]. The apoptosis of endothelial cells significantly reduces endothelial function and creates conditions for the development of atherosclerosis. Ox-LDL is one of the most important factors contributing to endothelial cell apoptosis. In vitro experiments have shown that an increased concentration of PCSK9 significantly accelerates this process (Figure 4). In patients with ST-elevation myocardial infarction (STEMI), all who were previously treated with statins, the level of apoptosis of human umbilical vein endothelial cells (HUVEC) that were treated with the patients’ serum was related to PCSK9 concentration [70]. They showed that ox-LDL-induced HUVECs apoptosis could be inhibited by PCSK9 siRNA. The exact mechanism by which the PCSK9i improved the endothelial function remains unknown. It is possible that its effect was mostly mediated by the reduction in LDL cholesterol levels; nevertheless, other mechanisms can perform the same role. Lipid-lowering treatments ameliorate oxidative stress and improve endothelial nitric oxide synthesis [71]. In addition, the inhibition of PCSK9 induces the upregulation of LDLR. This may increase the LDL cholesterol binding affinity of the LDLR, leading to an improvement in endothelial function [72]. On the other hand, PCSK9 is associated with a macrophage-mediated inflammatory response, and treatment with PCSK9 monoclonal antibodies leads to decreased migratory capacity and a reduced inflammatory response [73]. PCSK9 monoclonal antibodies attenuate the pro-inflammatory activation of endothelial cells and reduce the apoptosis of endothelial cells, smooth muscle cells and macrophages [24]. In addition to that, PCSK9 monoclonal antibodies may have an effect on circulating endothelial progenitor cells (cEPCs). The latter are characterized by positivity for CD34, CD133 and vascular endothelial growth factor receptor-2 (VEGFR-2), and they are involved in vascular repair as a response to endothelial injury [25]. In patients with increased cardiovascular risk, Maulucci et al. showed that after 2 months of treatment with evolocumab, the improved endothelial function was proportional to the decreased LDL cholesterol levels. The most-likely mechanism of action suggested is the inhibition of PCSK9-mediated LDL receptor degradation and recycling of LDL receptors back to the hepatic cell surface, which lowers serum LDL cholesterol levels [22]. In a small clinical study, Leucker et al. [26] reported that in HIV patients with near-optimal or above-the-goal mean LDL cholesterol values, as well as in HIV patients with other dyslipidaemias, PCSK9 inhibition with evolocumab significantly improved coronary endothelial function after 6 weeks of treatment. In our research, we found that the improvement in endothelial function during treatment with PCSK9i was also significantly dependent on the accompanying risk factors, primarily, smoking and diabetes [27].

Functional changes in the arterial vessel wall are followed by the morphological ones, which can be measured as an increased stiffness of the arterial wall or as an increased thickness of the intima-media of the carotid arteries (c-IMT). The link between serum cholesterol level and arterial stiffness may be explained by several potential mechanisms, the most important of which is likely the development of atherosclerosis, which has been consistently associated with increased arterial stiffness in subjects with and without severe hypercholesterolemia. However, cholesterol and, especially, ox-LDL have further non-atheromatous effects on the arterial wall, leading to stiffening. Ox-LDL promotes peroxynitrite formation and increased oxidative stress, which may lead to direct damage to elastin, the main elastic element of the arterial wall [28]. Furthermore, ox-LDL has pro-inflammatory effects characterized by increased serum levels of C-reactive protein (CRP), which has been associated with arterial stiffness in apparently healthy individuals [29]. Suitable non-invasive methods in early stages of atherosclerosis and related artery wall disorders include measuring arterial stiffness. Functional parameters that are independent predictors of cardiovascular disease include applied pulse wave velocity (PWV) and the augmentation index (Aix), both of which are suitable for screening and monitoring the efficiency of treatment [74]. PCSK9 was found to be independent predictor of PWV in 401 patients with type 2 diabetes who had not experienced previous cardiovascular events, even after adjusting for other cardiovascular risk factors [75]. In a recent study, the effect of a six-month add-on of PCSK9i on circulating PCSK9 and PWV was detected in a cohort of FH subjects. The PCSK9 plasma level was correlated with PWV levels at baseline. A reduction in PCSK9 plasma level appeared to be associated with a significant mechanical vascular improvement after PCSK9i therapy [76].

As with PWV, increased PCSK9 concentration is an independent predictive factor for increased c-IMT. Here, it is very important that the results adjusted to standard risk factors are also adjusted to the statin treatments, which increase the value of PCSK9. The development of carotid atherosclerosis may not only be dependent on the concentration of PCSK9 but also on the single nucleotide polymorphisms (SNPs) that affect PCSK9. A study that monitored approximately 1000 apparently healthy patients with an average age of just over 40 years for nearly 20 years showed that in addition to the concentration of PCSK9, one of the SNPs which determines the concentration of PCSK9 is important for the development of carotid atherosclerosis. Of the 26 SNPs that were studied, it was discovered that three (rs540796, rs562556 and rs631220) were associated with the presence of plaques after 20 years. Minor alleles of these 3 SNPs were also associated with LDL cholesterol and PCSK9 levels, both at the time of induction and at the end of the observation period. These three SNPs were highly linked, and one of them (rs562556) carried a missense gain-of-function mutation. Subjects who carried the minor allele of this SNP had higher LDL cholesterol levels and they more often developed carotid plaques. However, no correlations were found between the PCSK9 polymorphisms and PWV or c-IMT [77].

The extent of the atherosclerotic plaque is also related to the PCSK9 concentration. In 581 patients in whom coronary angiography was performed due to acute coronary syndrome, intravascular ultrasound virtual histology was also performed on a non-culprit lesion. The results showed that higher serum PCSK9 levels were associated with higher necrotic core fractions. This association was independent of the serum LDL cholesterol levels and statin treatments [78]. The global assessment of plaque regression with a PCSK9 antibody measured by intravascular ultrasound (GLAGOV) trial demonstrated that the addition of evolocumab to the treatment of patients with CAD who were pre-treated with statins had a favourable effect on the progression of coronary atherosclerosis as measured by intravascular ultrasound (IVUS) [79]. These results were further confirmed by the high-resolution assessment of coronary plaques in a global evolocumab randomized study (HUYGENS), which showed that the evolocumab treatment increased the stability of the atherosclerotic bed by reducing the lipid core and increasing the fibrous cap’s thickness [42].

## 5. Conclusions

Despite the fact that PCSK9 is one of the most important molecules in the metabolism of LDL cholesterol, its effects on the atherosclerosis process go far beyond regulating the concentration of LDL cholesterol. Although the primary and largest sources of PCSK9 are hepatocytes, we should not neglect the production of PCSK9 in endothelial cells, smooth muscle cells and, especially, macrophages. We should note that the increased production of PCSK9 in these cells is observed primarily in people with atherosclerotic changes, and that PCSK9 itself accelerates the activation of these cells and further stimulates the production of PCSK9. Hence, it is a vicious circle. It is likely that in the near future, determining the concentration of PCSK9 will be as self-evident as measuring the concentration of LDL cholesterol and Lp(a) is now. It would also be interesting to know whether PCSK9i and inclisiran reduce PCSK9 levels to the same extent. Perhaps the results of the ORION-3 study, where, in the first part, the patients received either evolocumab or inclisiran, can give us an answer, anticipating, of course, that the concentration of PCSK9 will be determined. Thus far, we only have the results for the group with inclisiran in terms of LDL cholesterol reduction [80]. However, is not entirely clear why the results of the group of patients who received evolocumab have not yet been revealed.

## Figures and Tables

**Figure 1 ijms-24-01966-f001:**
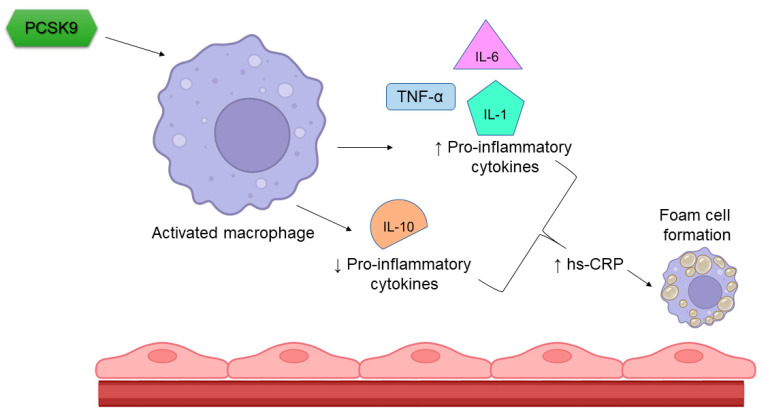
Influence of PCSK9 on inflammation. PCSK9 induces the expression of pro-inflammatory markers, decreases the expression of anti-inflammatory cytokines and promotes foam cell formation, all of which are the core elements of endothelial cells dysfunction and the progression of the atherosclerotic process. PCSK9—proprotein convertase subtilisin/kexin type 9, IL—interleukin, TNF-α—tumor necrosis factor α, hs-CRP—high-sensitivity C reactive protein.

**Figure 2 ijms-24-01966-f002:**
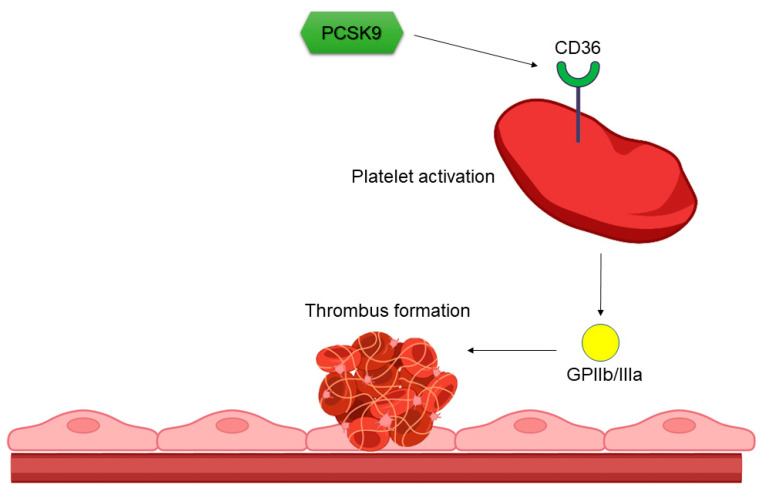
Influence of PCSK9 on platelet function. By binding to the CD36 receptor on a platelet’s surface, PCSK9 initiates a cascade reaction, resulting in GPIIb/IIIa formation, which enables increased platelet aggregation. PCSK9—proprotein convertase subtilisin/kexin type 9, CD36—cluster of differentiation, GPIIb/IIIa—glycoprotein IIb/IIIa.

**Figure 3 ijms-24-01966-f003:**
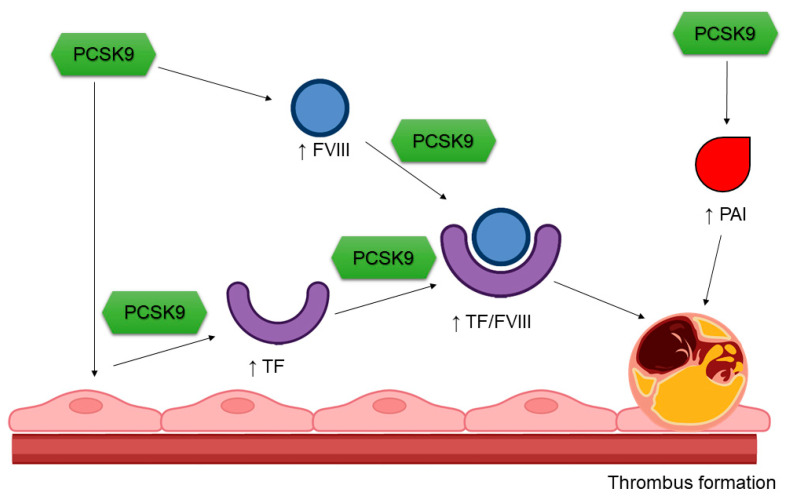
Influence of PCSK9 on coagulation and fibrinolysis. PCSK9 increases the release of TF from the endothelial cells, activates the extrinsic pathway of coagulation, increases the concentration of FVIII and, thus, increases the TF/FVIII complex. On the other hand, it increases the release of PAI-1 and reduces fibrinolytic capacity. PCSK9—proprotein convertase subtilisin/kexin type 9, TF—tissue factor, FVIII—factor VIII, PAI-1—plasminogen activator inhibitor-1.

**Figure 4 ijms-24-01966-f004:**
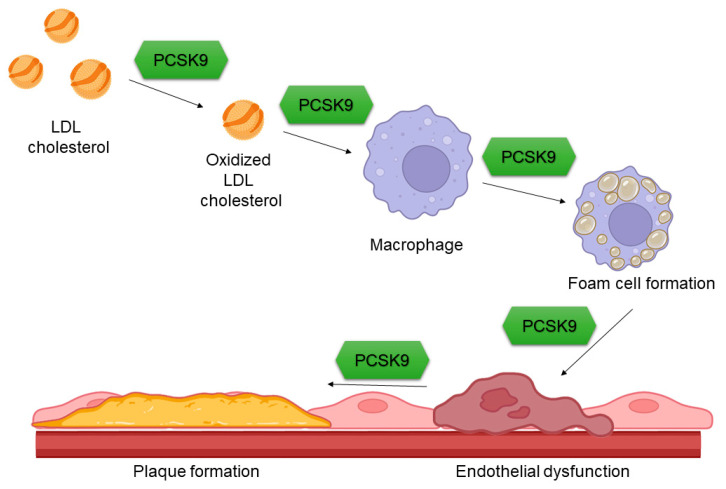
Influence of PCSK9 on arterial wall properties. PCSK9 accelerates the oxidation of LDL cholesterol and the transformation of macrophages into foam cells, resulting in endothelial dysfunction and, consequently, the development of atherosclerotic plaques. PCSK9—proprotein convertase subtilisin/kexin type 9, LDL—low density lipoprotein.

**Table 1 ijms-24-01966-t001:** Overview of the clinical studies that have evaluated the effects of PCSK9 on atherosclerotic risk factors.

Risk Factor	Study Population		Primary Endpoint	Outcome	Reference
	Description	n (M/F)			
Inflammation	Cross-sectional stable CAD	219	PCSK9 correlation with hs-CRP and fibrinogen	r = 0.153; *p* = 0.023 r = 0.211; *p* = 0.002	[10]
	Cohort of CAD-free participants	643	Carotid plaque formation	PCSK9 levels were significantly associated with new plaque formation after adjusting for LDL-C levels and other risk factors (relative risk per quartile increase 1.09, 95% CI 1.03–1.15, *p* = 0.008)	[11]
	Prospective cohort of patients with ACS	2030 (1501/529)	Association between PCSK9 and inflammation in the acute phase (hs-CRP)	*p* = 0.006	[8]
			Association between PCSK9 tertiles and all-cause death	*p* = 0.339	
	Randomized double-blind placebo-controlled study of patients with elevated Lp(a), with or without CAD	129 (60/69)	Altering of arterial-wall inflammation (18F-FDG PET/CT)	Week 16 index vessel MDS TBR was not significantly altered with evolocumab; (−8.3%) vs. placebo (−5.3%), *p* = 0.18	[12]
	Double-blind, patients with increased CV risk (atherosclerotic disease or familial hypercholesterolemia, LDL-C of > 100 mg/dL, not receiving statins)	50 (16/34)	Impact of alirocumab on arterial-wall inflammation (MDS TBR of the index carotid)	Significant decrease in MDS TBR of the index carotid (−6.1%; 95% CI −0.33 to −0.01; *p* = 0.04) compared to placebo (2.1%; 95% CI −0.09 to 0.15, *p* = 0.60)	[13]
Platelet function	Prospective, observational study of patients with ACS receiving prasugrel, ticagrelor or P2Y12 inhibitors and undergoing PCI	178	MACEs, association of PCSK9 with higher platelet reactivity	Direct association was found between increased PCSK9 serum levels and platelet reactivity (r = 0.30; *p* = 0.004); at one-year follow-up, PCSK9 was independently associated with increased ischemic MACEs, and the hazard ratio for upper vs. lower PCSK9-level tertile was 2.62 (95% CI 1.24–5.52; *p* = 0.01)	[14]
	Patients with symptomatic CAD	707 (500/207)	Association of platelet-derived PCSK9 and platelet aggregation	PCSK9i significantly reduced platelet-dependent thrombus formation (*p* < 0.05)	[15]
	Multicentre before-and-after study, in vitro, HeFH patients receiving statin +/− ezetimibe	80 (44/36)	Effect of plasma from HeFH patients on platelet activation in washed platelets before and after PSCK9i	PCSK9i reduced the serum levels of LDL-c, ox-LDL, thromboxane B2, sNOX2-dp and PCSK9 (*p* < 0.001)	[16]
	12-month follow up of patients with primary hypercholesterolemia, all receiving statin and 17 receiving ASA	24	Evaluation of platelet function parameters at baseline up to 12 months of treatment with alirocumab or evolocumab	Significant decrease in platelet aggregation in ASA HC patients (*p* < 0.0001) and significant decrease in platelet membrane expression of CD62P and plasma levels of the in vivo platelet activation markers in all HC patients	[17]
Coagulation and fibrinolysis	Short-term, non-randomized, controlled study of individuals with isolated hypercholesterolemia and atherosclerosis	21 (14/7)	Effect of alirocumab on a reduction in plasma levels/activity of fibrinogen, factor VIII and PAI-1	from 3.6 +/− 0.5 to 2.9 +/− 0.4 g/L, *p* < 0.001 from 143.8 +/− 16.7 to 114.5 +/− 14.1%, *p* < 0.001 from 74.9 +/− 13.9 to 52.8 +/− 9.1 ng/mL, *p* < 0.001	[18]
	Prospective cohort study of patients with angina-like chest pain and without lipid-lowering drugs	2293 (1387/906)	Association between PCSK9 concentration, routine coagulation indicators and MACEs	Patients with high PCSK9 levels had lower PT and APTT levels (*p* < 0.05) 186 (8.1%); MACEs occurred, and patients with high PCSK9 and low PT had higher incidence of MACEs	[19]
	Individuals with FH treated with statins alone or statin +/− ezetimibe	80	Correlation between PCSK9 and increased levels of TC, LDL-C, triglycerides and TGA	Inverse correlation between PCSK9 and peak (lowTF) (r = −0.352; *p* = 0.001) and positive correlation between PCSK9 and peak (lowTF) (r = 0.414; *p* = 0.001)	[20]
	Statin-intolerant patients with FH	30 (13/17)	Change in D-dimer and fibrinogen levels after start of evolocumab or alirocumab	Baseline median D-dimer levels of 0.34 mg/L vs. follow-up of 0.31 mg/L (*p* = 0.37) and baseline median fibrinogen levels of 3.2 g/L vs. follow-up of 3.4 g/L (*p* = 0.38)	[21]
Arterial wall properties	Patients with previous myocardial infarction before and after 2 months of treatment with evolocumab 140 mg twice per month	14	Brachial artery vasoreactivity test -increased brachial artery diameter -increased velocity time integral	*p* = 0.001 *p* = 0.045	[22]
	Single-centre study of people living with HIV (PLWH) and dyslipidaemia	30 (24/6)	Effect of evolocumab on changes in coronary cross-sectional area and coronary blood flow during isometric handgrip exercise	+5.6+/−5.5% in PLWH group and +4.5+/−3.1% in dyslipidaemia group, *p* < 0.01	[23]
	Cross-sectional analysis of Caucasian patients with type 2 diabetes	401 (241/160)	Correlation between PCSK9 and arterial stiffness (carotid-femoral pulse wave velocity)	PWV resulted directly and was significantly correlated with PCSK9 circulating levels (r = 0.408, *p* = 0.003)	[24]
	HeFH subjects	26 (17/9)	Impact of PCSK9 plasma levels on pulse wave velocity	r = 0.411 *p* = 0.001	[25]
	Longitudinal familial cohort from the Lorraine region of France	997 (509/488)	Association between PCSK9 levels and carotid arterial plaques	Odds ratio of 2.14; 95% CI = 1.28–3.58; *p* < 0.05	[26]
	Patients undergoing coronary angiography for ACS or stable angina	581	Association between serum PCSK9 levels and fraction of plaque of necrotic core tissue	1.24% increase per 100 µg/L increase in PCSK9, *p* = 0.001	[27]
	Multicentre, double-blind, placebo-controlled randomized clinical trial of patients with angiographic coronary disease	968 (696/272)	Change in percent atheroma volume from baseline to week 78 measured by IVUS and change in normalized total atheroma volume	0.005% increase with placebo, 0.95% decrease with evolocumab, *p* < 0.001 0.9 mm^3^ decrease with placebo and 5.8 mm^3^ decrease with evolocumab, *p* < 0.001	[28]
	Double-blind, placebo-controlled trial of patients with non-ST-elevation myocardial infarction treated with evolocumab or placebo	161 (115/46)	Increase in minimum fibrous cap thickness, decrease in maximum lipid arc and decrease in macrophage index throughout the arterial segment	+42.7 vs. +21.5 µm, *p* = 0.0015; −57.5° vs. −31.4°, *p* = 0.04; and −3.17 vs. −1.45, *p* = 0.04	[29]

M/F—male vs. female ratio, ACS—acute coronary syndrome, APTT—activated partial thromboplastin time, ASA—acetyl salicylic acid, CAD—coronary artery disease, CI—confidence interval, CRP—C-reactive protein, CV—cardiovascular, HeFH—heterozygous familial hypercholesterolemia, IVUS—intravascular ultrasound, LDL-C—low density lipoprotein cholesterol, MACEs—major adverse cardiovascular events, MDS—most-diseased segment, ox-LDL—oxidized low-density lipoprotein, PCSK9—proprotein convertase subtilisin/kexin type 9, PCSK9i—proprotein convertase subtilisin/kexin type 9 inhibitor, sNOX2-dp—soluble-NADPH oxidase 2-derived peptide, PAI-1—plasminogen activator inhibitor-1, PCI—percutaneous coronary intervention, PT—prothrombin time, PWV—pulse wave velocity, TBR—target-to-background ratio, TC—total cholesterol, TF—tissue factor, TGA—thrombin generation assay, 18F-FDG PET/CT—fluorine-18 fluorodeoxyglucose positron emission tomography/computed tomography.

## Data Availability

Not applicable.

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
