# Peer review of "PCSK9 as an Atherothrombotic Risk Factor"

_ijms, 2023, doi:10.3390/ijms24031966_

Round 1

Reviewer 1 Report

Dear Editor,

I carefully read the article "PCSK9 as atherothrombotic risk factor" by Sotler and Šebeštjen.

My comments and suggestions for the authors are the following:

 - English language needs to be carefully revised and improved.

 - Table 1: The authors should include in the table also information regarding the prevalence by sex in the clinical studies.

 - In the title of the manuscript, the authors should specify that this is a state-of-the-art (or a narrative) review.

 - In the manuscript, the authors should specify which criteria they followed in order to select the articles to include among the references.

 - In their manuscript, the authors should certainly refer to the most comprehensive meta-analysis on inclisiran: https://doi.org/10.1016/j.ahjo.2022.100127. Moreover, they should discuss also doi: 10.1016/j.jacl.2022.10.009, doi: 10.1016/j.biopha.2022.113957 and doi: 10.1080/14740338.2022.1988568.

Reviewer 2 Report

 I have the opportunity to review the paper entitled “PCSK9 as atherothrombotic risk factor” written by Sotler and  Šebeštjen. It is a well-written paper which summarizes the current knowledge on the effects of PCSK9 and its inhibition beyond reduction of LDL particles in cardiovascular disease.

 COMMENTS

I’ll focus my review in the role of PCSK9 on inflammation, which can be seen as a systemic (hsCRP levels) o local response. Contrary to we expected, the marked reduction in LDL cholesterol seen with aliro- and evolocumab did not reduce the hsCRP levels in clinical trials. This markedly differed from the reduction of LDL cholesterol using statins (but also bempedoic acid). The theory said that a marked reduction in LDL cholesterol induce a depletion of LDL particle in the arterial wall, so less ox-LDL is produced and less macrophages are activated: hsCRP represent the low-grade inflammation in the arterial wall. However, despite of both aliro- and evolocumab have shown marked reduction in size of the atherosclerotic plaques (Nicholls et al., 2022; Räber et al., 2022), nothing happens in terms of hsCRP. In other words, only drugs that reduce cholesterol synthesis (BA and statins) are able to decrease hsCRP levels.

As mentioned by the authors, inflammation in the arterial wall can be measured by 18-FDG PET/CT. Again, there is controversial data. Among the three papers (in my knowledge), published on PCSK9i and arterial wall inflammation measured as PET/CT, in two of them there was a significant reduction in the uptake of the radiopharmaceutical (Hoogeveen et al., 2019; Vlachopoulos et al., 2019), but not in the third (Stiekema et al., 2019).  I suggest to add the paper of Vlachopoulos to the references and perhaps change the paragraph (lines 150-155) to indicate that elevated Lp(a) may be responsible por the persistence of elevated inflammation.

A recent paper and an editorial indicate that PCSK9i increased LPS uptake by endothelial cells (and also hepatocytes) promoting inflammation at local level (https://doi.org/10.1016/j.atherosclerosis.2022.11.003). I suggest to cite them.

Concerning the association between PCSK9 and coagulation/fibrinolysis, perhaps  the authors would like  to mention the controversial results of sub analysis of the Fourier and Outcomes studies on the risk of deep vein thrombosis and venous thromboembolism (Marston et al., 2020; Schwartz et al., 2020).

Hoogeveen, R. M., T. S. J. Opstal, Y. Kaiser, L. C. A. Stiekema, J. Kroon, R. J. J. Knol, W. A. Bax, H. J. Verberne, J. H. Cornel, and E. S. G. Stroes, 2019, PCSK9 Antibody Alirocumab Attenuates Arterial Wall Inflammation Without Changes in Circulating Inflammatory Markers: JACC. Cardiovascular imaging, v. 12, no. 12, p. 2571–2573, doi:10.1016/J.JCMG.2019.06.022.

Marston, N. A. et al., 2020, The Effect of PCSK9 (Proprotein Convertase Subtilisin/Kexin Type 9) Inhibition on the Risk of Venous Thromboembolism: Circulation, v. 141, no. 20, p. 1600–1607, doi:10.1161/CIRCULATIONAHA.120.046397.

Nicholls, S. J. et al., 2022, Effect of Evolocumab on Coronary Plaque Phenotype and Burden in Statin-Treated Patients Following Myocardial Infarction: Cardiovascular Imaging, v. 15, no. 7, p. 1308–1321, doi:10.1016/J.JCMG.2022.03.002.

Räber, L. et al., 2022, Effect of Alirocumab Added to High-Intensity Statin Therapy on Coronary Atherosclerosis in Patients With Acute Myocardial Infarction: The PACMAN-AMI Randomized Clinical Trial: JAMA, v. 327, no. 18, p. 1771–1781, doi:10.1001/JAMA.2022.5218.

Schwartz, G. G. et al., 2020, Peripheral Artery Disease and Venous Thromboembolic Events After Acute Coronary Syndrome: Role of Lipoprotein(a) and Modification by Alirocumab: Prespecified Analysis of the ODYSSEY OUTCOMES Randomized Clinical Trial: Circulation, v. 141, no. 20, p. 1608–1617, doi:10.1161/CIRCULATIONAHA.120.046524.

Stiekema, L. C. A., E. S. G. Stroes, S. L. Verweij, H. Kassahun, L. Chen, S. M. Wasserman, M. S. Sabatine, V. Mani, and Z. A. Fayad, 2019, Persistent arterial wall inflammation in patients with elevated lipoprotein(a) despite strong low-density lipoprotein cholesterol reduction by proprotein convertase subtilisin/kexin type 9 antibody treatment: European heart journal, v. 40, no. 33, p. 2775–2781, doi:10.1093/EURHEARTJ/EHY862.

Vlachopoulos, C. et al., 2019, Long-Term Administration of Proprotein Convertase Subtilisin/Kexin Type 9 Inhibitors Reduces Arterial FDG Uptake: JACC: Cardiovascular Imaging, v. 12, no. 12, p. 2573–2574, doi:10.1016/J.JCMG.2019.09.024.
